# Inhibition of Q235 Carbon Steel by Calcium Lignosulfonate and Sodium Molybdate in Carbonated Concrete Pore Solution

**DOI:** 10.3390/molecules24030518

**Published:** 2019-01-31

**Authors:** Bing Lin, Yu Zuo

**Affiliations:** Beijing Key Laboratory of Electrochemical Process and Technology for Materials, Beijing University of Chemical Technology, Beijing 100029, China; 2015400068@mail.buct.edu.cn

**Keywords:** carbonated concrete environment, carbon steel, calcium lignosulfonate (CLS), sodium molybdate, synergistic effect, corrosion experiments, inhibition

## Abstract

The inhibition effect and mechanism of a compound calcium lignosulfonate (CLS) and sodium molybdate inhibitors for Q235 carbon steel in simulated carbonated concrete pore solution (pH 11.5) with 0.02 mol/L NaCl are studied using electrochemical and surface analysis techniques. The results show that in carbonated simulated concrete pore (SCP) solution CLS and Na_2_MoO_4_ show a synergistic inhibition effect. The compound inhibitor can be defined as mix-type inhibitor. With 400 ppm CLS plus 600 ppm Na_2_MoO_4_, the pitting potential moves positively about 200 mV, and the inhibition efficiency reaches 92.67%. After 24 h immersion, the IE% further increases up to 99.2%. The surface analysis results show that Na_2_MoO_4_ could promote stability of the passive film, and the insoluble molybdenum compounds and CaO/Ca(OH)_2_, together with adsorbed CLS, deposit on the steel surface, forming a complex film. The compounded film effectively inhibits corrosion of the steel.

## 1. Introduction

Reinforced concrete is the most widely used construction material [1,2,3], while corrosion of reinforcing steel is the main reason leading to the premature deterioration of reinforced concretes [4,5]. Corrosion of reinforcing steel is related to the carbonation of concrete [2,3,6] and the presence of chloride ions [7,8]. The carbonation of concrete is the result of CO_2_ diffusion and reaction with the hydrated product (Ca(OH)_2_), which leads to the composition change of the concrete pore solution and the decrease of pH value. Several techniques have been employed to reduce corrosion of reinforcing steel, among which corrosion inhibitor is the most practical method owing to its advantages of low cost, high efficiency, and wide applicability [1,2,3,9,10,11]. For example, Ca(NO_2_)_2_ is an inorganic corrosion inhibitor commonly used in reinforced concrete construction [12,13,14]. NO_2_^−^ could help to form a uniform passive film on a steel surface [13,14] and prevent the steel from corroding. However, most of the commercial inhibitors remain unfriendly to the environment. In recent years, researchers have paid considerable attention towards the development of various more effective and greener inhibitors to prevent the corrosion of rebars [15,16].

Lignosulfonates (LS) are the by-products of the pulping waste liquor in acid sulfite pulp mills [17]. One of the most common applications of lignosulfonates is as dispersing agents and set-retarding agents in concretes [18]. As corrosion inhibitors, lignin-based compounds contain plenty of hydroxyl, carboxyl, or methoxyl groups, which can be adsorbed onto a metal surface by sharing their lone pair electron or π-electron with the free d-orbital of metal. In acidic solution, the inhibition efficiency of lignin and its modifications for steel depend on the number of carboxy groups in the macromolecule [19], the inhibitor concentration, and the environmental factors [20]. In neutral aqueous solutions with Cl^−^, lignin monomers acted as mixed type inhibitors on steel corrosion [21] and the inhibitive performance of lignin derivatives depended on their functional groups [22]. Bishop et al. [23] have confirmed that the –SO_3_ and –OH groups present in LS can be adsorbed on a steel surface and generate an impermeable structure. Li et al. [24] revealed that LS adsorbed in the form of aggregates and the adsorption isotherm follows the Freundlich model at low pH values. In neutral solutions, LS increasingly adsorbed on steel in the form of well separated single molecules, which can be well fitted using the Langmuir equation. However, there is little research involving calcium lignosulfonate (CLS) as a corrosion inhibitor in alkaline solution. Our work [25] showed that CLS is a good corrosion inhibitor in saturated Ca(OH)_2_ solutions with Cl^−^ for carbon steel. The adsorption film on a steel surface is formed by both physisorption and chemisorption. Particularly, CLS could be preferentially adsorbed on the active sites on a steel surface using the sinapyl alcohol group, which is beneficial to the inhibition of pitting corrosion.

Molybdate is an environment-friendly anodic passivation inhibitor [26,27]. The inhibition mechanism of molybdates has been studied by many scholars, and competitive adsorption [28], adsorption film [27,29], and oxidation [30] are the dominant views. In order to enhance the inhibition effective of molybdates, some authors [31,32,33] tried to associate Na_2_MoO_4_ with organic inhibitors. It was reported [33] that the synergistic effect is present between molybdate and benzotriazole (BTA). The compounded inhibitor could promote the transformation of FeOOH and Fe_2_O_3_ in the passive film, and both passivation and pitting corrosion resistance were promoted.

On the other hand, most studies focusing on inhibitors for reinforcing steel were performed with high pH concrete pore solutions. The carbonation of concretes would directly affect the passivation film on reinforcing steel surfaces, but few studies were carried out in carbonated concrete environments. In the present work, the combined inhibition effect of CLS and molybdate on Q235 carbon steel in simulated carbonated concrete pore solution is studied with electrochemical measurements and surface analysis methods. Particularly, the synergistic effect between Na_2_MoO_4_ and CLS is paid attention to.

## 2. Results and Discussion

### 2.1. Inhibition Evaluation

#### 2.1.1. Potentiodynamic Polarization Measurement

Figure 1a shows the polarization curves of Q235 carbon steel after immersion in test solutions without and with various concentrations of CLS. The polarization curves show typical passivation-pitting corrosion behavior. Figure 1b shows the electrochemical parameters obtained from the polarization curves.

The corrosion potential (*E_corr_*) of carbon steel in the test solution is about −290 mV_SCE_, and almost has almost no change as CLS concentration increases, indicating CLS acts as a mixed-type inhibitor [34,35]. The inhibition efficiency (IE%) increases as the CLS concentration increases, which is consistent with the results of other authors [17,25]. As an organic inhibitor, CLS could form an adsorption film on a steel surface [25], which has a barrier effect to prevent general corrosion. The pitting potential (*E_b_*) is the potential at which the anodic current increases rapidly [36], and the localized corrosion starts above this potential. As the CLS concentration increases from 100 to 800 ppm, the E_b_ value increases from -146 mV_SCE_ to 57 mV_SCE_, and then slightly decreases. This result indicates that the pitting corrosion susceptibility of carbon steel decreases as the inhibitor concentration increases. The max current density (*i_peak_*), which is related to the growth rate of the corroded pits [37], remains almost unchanged as the CLS concentration increases. The repassivation potential (*E_pp_*) is the intersection potential of the forward and reverse scans, and the steel potential must be above *E_pp_* for existing areas of localized corrosion to propagate [38]. As the CLS concentration increases the *E_pp_* shows relatively smaller shifts to positive, compared with the obvious shifts of the E_b_ values. The results of *i_max_* and *E_pp_* reveal that, as a corrosion inhibitor, CLS has little effect on the repassivation process of carbon steel. Compared with the inhibition effect of CLS in pH 12.5 concrete pore solution [25], the inhibition ability of CLS in carbonated SCP solution decreases for both general corrosion and localized corrosion. Therefore, the combination of CLS with other inhibitors to improve the inhibition effect is necessary.

Figure 2 shows the polarization curves and the electrochemical parameters of Q235 steel in test solution with various concentrations of Na_2_MoO_4_. The *E_corr_* slightly increases with the increase of MoO_4_^2−^ concentration, indicating that Na_2_MoO_4_ acts as a mixed-type inhibitor [34,35] predominantly with anodic effectiveness [39]. Na_2_MoO_4_ is a moderate inhibitor for general corrosion, and the IE% increases as the Na_2_MoO_4_ concentration increases. The passive current density decreases obviously with the increase of the inhibitor concentration, indicating Na_2_MoO_4_ promoted passivation of the steel. Fe_2_(MoO_4_)_3_ complex is insoluble and protective in neutral and alkaline media [29], which could enhance the protection film formed on steel surfaces [39]_._ Refaey et al. [40] reported that the formation of a protective film played a critical role on the inhibition effect of molybdate. The increased E_b_ and passivation region (*E_b_*-*E_corr_*) due to the increase of MoO_4_^2−^ concentration might be associated with the effect of molybdate reducing the number and magnitude of metastable pitting transients [39,41]. In other words, molybdate ions could block the active sites on the surface [42] and affect the nucleation of pits by deactivating or reducing the number of sites, leading to decreased pitting corrosion susceptibility [43]. As shown in Figure 3, on the reverse scan, a potential step, the pit transition potential (*E_ptp_*) [39,44,45], is detected for steel in the test solution with molybdate. The *E_ptp_* is a characteristic potential that is correlated with repassivation at the pit bottom [44,45], which might lead to the concentration gradients for mass transport and promote further pit nucleation. In the pit environment, a series of chemical reactions involving hydrolysis and polymerization of molybdates may occur as the pH value decreases [29,46]:7MoO_4_^2−^+8H^+^→Mo_7_O_24_^6−^+4H_2_O(1)

Mo_7_O_24_^−^ has a chelate effect with iron(III) to form complexes, which could help repassivation of the pit. In Figure 2b, the *E_ptp_* potential increases as the molybdate concentration increases, which means that molybdate with relatively higher concentration could help to form a repassivation film at the pit bottom.

The mechanism of molybdate inhibition in carbonation SCP solution could be inferred. In the first step, the MoO_4_^2−^ ions competitively adsorb on the steel surface with Cl^−^ ions [47,48]. Then the passivation film could be enhanced by the adsorbed molybdate ions [49] and a precipitation film composed of oxidized molybdenum forms on the steel surface. The composite film could increase both the general corrosion and pitting corrosion resistances. However, once the pits occur, MoO_4_^2−^ cannot stop the pit growth and self-catalyzed corrosion occurs inside the pits which accelerates the growth of pits.

Figure 3 shows the cyclic potentiodynamic polarization (CPP) curves of Q235 carbon steel in carbonated SCP solution with different ratios of CLS and Na_2_MoO_4_ (total 1000 ppm), and the electrochemical parameters are shown in Table 1. The *E_corr_* slightly increases as the CLS ratio increases, which is the same as the result of molybdate compound with glycol [47]. The compound inhibitor acts as a mix-type inhibitor with predominantly anodic effectiveness [47]. The IE% slightly decreases as the CLS ratio increases. The synergistic parameter (S), which reveals the interaction relationship between CLS and Na_2_MoO_4_, is calculated using the following equation [50,51]:(2)$$S=\frac{1-{\mathrm{IE}\%}_{\left(1+2\right)}}{1-{\mathrm{IE}\%}_{c}}$$
where IE%_(1 + 2)_ = (IE%_1_ + IE%_2_) − (IE%_1_ × IE%_2_), and IE%_1_, IE%_2_, and IE%_c_ are the inhibition efficiencies for CLS, Na_2_MoO_4_, and CLS/Na_2_MoO_4_ compound, respectively. The calculated S values are all above 1, indicating the synergistic behavior of the selected inhibitor combination [50], and the S value increases as the CLS concentration decreases, which means the synergistic effect between the two inhibitors decreases. The obtained *E_b_* potentials for each compound inhibitor ratios are relatively close and obviously higher than the value without inhibitor. The compound with 400 ppm CLS and 600 ppm Na_2_MoO_4_ shows the highest *E_b_*. The *E_pp_* significantly increases as the CLS ratio decreases, which means that a decrease of the CLS ratio in the compound inhibitor could promote the repassivation of carbon steel, while adding CLS or molybdate alone does not show this promoting effect. The difference between *E_b_* and *E_pp_* represents the repassivation tendency of pits on steel surface, which decrease as the molybdate concentration increases. This result suggests that the high ratio of molybdate in the compound inhibitor would be beneficial to inhibit localized corrosion. It has been reported that pitting corrosion could not be inhibited and might even be promoted if the molybdate concentration is too low [52,53]. This might be attributed to the pH changes in the pit environment in which the pH cannot be raised enough to passivate the pit [27,52]. The above results show that the compound inhibitor effectively increases both the general corrosion resistance and the pitting corrosion resistance, and the optimum ratio is 400 ppm CLS compound with 600 ppm Na_2_MoO_4_.

The pre-filming time has an important role in the inhibition processes [54,55]. CLS could form an adsorption film on a steel surface after 10 h pre-filming in concrete pore solution, which could reduce the content of Cl^−^ adsorbed on the steel surface and prevent the local enrichment of Cl^−^ [25]. As shown in Figure 4a, for the samples immersed in solution without inhibitor, with extended immersion time, *E_corr_* decreased and *i_corr_* increased obviously, and the passivation behavior disappeared. While for the samples with compound inhibitor, shown in Figure 4b, the polarization curves show typical passivation-pitting behavior. As the pre-filming time increases the *E_corr_* increases, the *i_corr_* decreases, and the IE% (calculated by *i_corr_*) increases from 92% to 99%. This result reveals that the compound inhibitor could enhance the general corrosion resistance after pre-filming time. *E_b_* for the inhibited steels obviously increases as the immersion time increases, while the *E_pp_* shows almost no shift, which suggests that in the carbonated solution the inhibitor could help Q235 steel maintain and improve the passivation state as the pre-filming time increases, especially for the first 16 h. The results confirm the good inhibition effect of the compound inhibitor after pre-filming time on both general corrosion and localized corrosion.

#### 2.1.2. Electrochemical Impedance Spectroscopy Measurements

To investigate the inhibitive behavior of CLS, Na_2_MoO_4_, and the compound inhibitor, a series of EIS tests were performed in carbonated SCP solution with various concentrations of different inhibitors. As shown in Figure 5, a single depressed capacitive semicircle is presented in all the Nyquist plots, and the capacitive semicircle radius increases as the inhibitor concentration increases. This result reveals that the addition of inhibitor did not change the mechanism of the corrosion process but inhibited corrosion by forming protective film on the steel surface [56]. The inhibitive system could be described using the model of the solution/steel interface, as shown in Figure 5d. In the equivalent circuit, R_s_ is the solution resistance and R_ct_ represents the charge-transfer resistance corresponding to the corrosion reaction at the metal substrate/solution interface. R_film_ and C_film_ represent the resistance and the capacitance of the protection film, respectively. The constant phase angle elements (CPE) is used in the model to compensate for the non-homogeneous electrode surface. The fitted parameters are shown in Figure 6.

Figure 5a shows the Nyquist plots of samples immersed in test solution with various concentrations of CLS. As the CLS concentration increases the diameter of the capacitive loop increases obviously. This phenomenon might be due to the increase of surface coverage by the adsorbed inhibitor film [56]. From the parameters in Figure 6, the R_ct_ increases and the R_film_ shows almost no shift as the CLS concentration increases, which suggests that the corrosion reaction was inhibited by the adsorption CLS film on the steel surface. The C_dl_ decreases as the CLS concentration increases, indicating that the status of the steel/solution interface has been changed by the adsorption of CLS. Furthermore, the C_film_ decreases with increased CLS ratio in the compound inhibitor. According to the Helmholtz model, the capacitance is inversely proportional to the surface changes [50,56]:C = ε_0_εS/d(3)
where d is the film thickness, S is the surface area of working electrode, ε_0_ is the permittivity of air and ε the local dielectric constant [56,57,58]. The decrease of C_dl_ and C_film_ might be attributed to the replacement of the adsorbed water molecules at the metal surface by CLS molecules, which have a lower dielectric constant. The adsorbed CSL film and the passive film could form a double-layer barrier on the steel surface [25]. As the CLS concentration increased the adsorbed CLS molecules increased, resulting in a dense and homogeneous inhibition film on the steel surface.

From Figure 5b and Figure 6, it can be seen that the C_dl_ decreases and the R_ct_ increases as the Na_2_MoO_4_ concentration increases, which means molybdate could increase the corrosion resistance to decrease the corrosion rate. As the molybdate concentration increases, the C_film_ value has almost no shift, and the R_film_ increases significantly. The EIS results confirm that a barrier layer, depending on the concentration of molybdate, blocked the adsorption of Cl^−^ [29]. In addition, molybdate could enhance the passivation film to protect the steel. Gong et al. [59] reported that molybdate enhances the self-healing ability of passive films, thereby leading to improvement of corrosion resistance.

After adding compound inhibitor, the Nyquist plots are shown in Figure 5c and the parameters are displayed in Figure 6. As the CLS ratio increases, the C_dl_ decreases slightly and the R_ct_ almost has no change, again indicating that the compound inhibitor could increase the corrosion resistance. The C_film_ value significantly decreases and the R_film_ value decreases slightly. In the compound inhibitor, the barrier effect of the protective film increases as the CLS ratio increases, and the passive film is enhanced to prevent corrosion as the molybdate ratio increases. The results revealed the compound inhibitor has a better inhibition effect for carbon steel in test solution.

In order to confirm the effect of pre-filming time on inhibition of the compound inhibitor, EIS measurements for Q235 carbon steel in the test solution with compound inhibitor for various immersion times were carried out and the results are shown in Figure 7. As the immersion time increases the diameter of the capacitive loop increases, which means as the pre-filming time increases the inhibition effect increases, which might be due to the increase in the surface coverage [56] or in the thickness of the protection film. From the EIS parameters shown in Figure 7b, the C_film_ decreases obviously and the R_film_ slightly increases as the immersion time increases, which also suggests the formation of adsorption film on the steel surface. In addition, the decreased C_dl_ and increased R_ct_ indicate that the corrosion reactions have been suppressed by the inhibitor. The above results confirm the results of the polarization measurements and suggest that after 16 h to 24 h pre-filming the compound inhibitor would show better inhibition effect.

### 2.2. Surface and Isotherm Adsorption Studies

#### 2.2.1. X-ray Photoelectron Spectroscopy Measurement and Contact Angle

In order to determine the chemical composition of the inhibited surface, XPS analyses were used to characterize the Q235 carbon steel after 1 h immersion in carbonated SCP solutions with 800 ppm CLS, 800 ppm Na_2_MoO_4_, and 400 ppm CLS, plus 600 ppm Na_2_MoO_4_, respectively. The obtained high-resolution peaks for Ca 2p, C 1s, O 1s, S 1s, Fe 2p, and Mo 3d core levels are fitted and shown in Figure 8, Figure 9 and Figure 10.

In Figure 8a, the C 1s spectra for CLS and compound inhibitor treated steel surface shows four peaks. The peak at 284.8 eV and 287.9 eV can be attributed to the C-C/C-H bonds [35,44,60] and the C-O bonds [50,61] in CLS molecules, respectively. The peak at 286.4 eV may be assigned to the carbon atom bonded to S in C-S bonds [60]. This peak on the compound inhibitor treated surface shifts negatively to 286 eV, suggesting that the adsorbed CLS may accept the feedback electrons from Fe atoms on the surface. The last peak at 292.2 eV can be attributed to the benzene ring in the organic inhibitor [60,61]. This peak shifts positively to 293.4 eV on the compound inhibitor treated surface, indicating the aromatic ring of the CLS supplied electrons to Fe atoms [62]. Figure 9 presents the Ca 2p and S 2p spectra for the CLS and compound inhibitor treated sample. Ca and S are the characteristic elements in CLS, which could confirm the adsorption of CLS. The Ca2p spectrum is decomposed into two peaks at 351 eV and 347.4 eV, which are attributed to CaO/Ca(OH)_2_ [25,63] and Ca-O-S bonds [25] on the steel surface, respectively. This result suggests the CaO/Ca(OH)_2_ has been precipitated on the steel surface and could enhance the adsorption of CLS by forming Ca-O-S bonds [25] to form a more complete film on the surface. For S 2p, there are two peaks. The first one is the peak at 168.4 eV [25,50] attributed to Ca-O-S, and the co-adsorption of CLS and Ca could protect the surface from corrosion [25,60,64]. On the compound inhibitor treated surface, this peak is widened, which might be due to the M-S-O bonds, where M could be Ca^2+^, Fe^2+^, or Fe^3+^. The area ratio of M-O-S on the compound inhibitor treated sample is much larger than that on the CLS treated sample, which means molybdate could enhance the adsorption of LS^2−^. The other peak at 167 eV [25,50] can be attributed to S in the organic inhibitor. The above XPS results confirm the adsorption of CLS and reveal the precipitation of oxides or hydroxides of Ca and Mo, which could help to enhance the adsorption of the organic inhibitor.

As shown in Figure 9, the high-resolution Mo 3d spectra are detected on both the surface treated using Na_2_MoO_4_ and the compound inhibitor. The two peaks at 231.9 eV is attributed to Mo^4+^, such as MoO_2_ [62], and the peak at 235.1 eV is the peak of Mo^6+^ [33], which could confirm the precipitation of MoO_x_ and adsorption of molybdate on the surface. The passive film formed by the reduction of molybdate mainly consists of iron oxide and molybdenum oxide. The reduction of Mo^6+^ could help to form a more stable film [47,65].

The O1s peaks for Q235 carbon steel immersed in test solutions with different inhibitors are shown in Figure 10a. The first peak at 530.1 eV corresponds to O^2−^, which could be attributed to the bond with Fe^3+^ in Fe_2_O_3_, Fe_3_O_4_, or MoO_4_^2−^ [33,35]. The second peak at 531.6 eV can be attributed to the metal oxides and hydroxides or H_2_O species on the steel surface [35,66]. The third peak may be assigned to the oxygen atoms of the O-C bond in the organic compounds from surface contamination and from the adsorbed water at 532.7 eV [60,62]. The deconvolution of the high-resolution Fe 2p3/2 spectra of the steel surface treated using different inhibitors show several peaks, which are shown in Figure 10b. The first component at 706.8 eV is attributed to metallic iron (Fe^0^) [35]. The peaks at 708.3 eV and 710.4 eV can be attributed to the Fe^2+^ present in FeO [35,39] and Fe_3_O_4_ [60], respectively. The peak at 711.5 eV is assigned to Fe^3+^, which can be associated to the ferric oxide/hydroxide species such as Fe_2_O_3_, Fe_3_O_4_, and FeOOH [50,60]. The peak at 714.2 eV is also assigned to Fe^3+^, which might be related to adsorption of the organic inhibitor [35,67]. In order to understand the effect of immersion time on the composition of passive film, XPS spectra of the samples immersed in the carbonated SCP solution with different inhibitors for different times were measured. The Fe^3+^/Fe^2+^ ratio could be used to evaluate the stability of the passive film on steel [29], as shown in Figure 11. As the immersion time increases, the Fe^3+^/Fe^2+^ ratio in the film obtained in solution without inhibitor decreases, indicating that the stability of the passive film decreases with time. For the steel treated using the CLS, the Fe^3+^/Fe^2+^ ratio slightly decreased in 30 h and then increased, which might be due to the adsorption of CLS replacing water molecules on the steel surface and reducing the passivation tendency. For the molybdate treated sample, the Fe^3+^/Fe^2+^ ratio increases as the immersion time increases, which confirms the previous result that molybdate could promote passivation of the surface. For the film formed in solution with the compound inhibitor, the Fe^3+^/Fe^2+^ ratio is the highest for different immersion times. This result also reveals that the combination of CLS and molybdate would further enhance the stability of the passive film.

The contact angle is an important parameter reflecting the wettability of materials [68,69]. The contact angles of Q235 steel surface treated using different inhibitors for 24 h are shown in Figure 12 and Table 2. The average contact angle for steel immersed in solution without inhibitor is 68.76°, which is consistent with references [68]. The inhibitor pretreatment induced significant changes in the contact angle. CLS causes the contact angle decrease obviously, which might be due to the fact that CLS is a surfactant and the adsorption film formed using CLS contains lots of methoxy, hydroxy, and other hydrophilic groups. The hydrophilic adsorption film could prevent the adsorption of aggressive anions and protect the steel surface [25]. Na_2_MoO_4_ could slightly increase the contact angle of the steel surface, which might be due to the Mo oxide precipitations decreasing the surface roughness [70]. The contact angle of the compound inhibitor treated steel is about 33.5°, which is between the values of the surface using single inhibitor treatment. The contact angle test results are consistent with the corrosion rate results and reveal the formation of the inhibition film on the steel surface.

#### 2.2.2. Inhibition Mechanism of CLS and Na_2_MoO_4_ Compound Inhibitor

The inhibition mechanism for the compounded inhibitor could be inferred using the above results, as shown in Figure 13. CLS molecules dissociate instantly to lignosulfonate (LS^2−^) ions and calcium ions (Ca^2+^). The steel surface immersed in the SCP solution has extra positive charges, providing a certain driving force for the adsorption of negative (LS^2−^, MoO_4_^2−^, Cl^−^) ions [25,71]. The inhibitor ions (MoO_4_^2−^ and LS^2−^) would adsorb on the steel surface competitively against Cl^−^ to protect the passive film. The adsorbed molybdate may react with the iron matrix to form Fe^2+^ and MoO_2_, and the Fe^2+^ would be further oxidized to form Fe^3+^. Fe^3+^ could help to form more stable passivation film [29,39] or react with molybdate to form insoluble Fe_2_(MoO_4_)_3_. The deposited MoO_2_ on steel surface would block the adsorption of Cl^−^ and prevent the steel from corrosion. Ca^2+^ in alkaline solution can form Ca(OH)_2_ or CaO, which could precipitate on steel surfaces. Furthermore, the insoluble molybdenum compounds, together with CaO/Ca(OH)_2_, could precipitate on the passive film to form a complex film, further enhancing the protection of the film. The precipitated Ca or Mo compounds could also enhance the adsorption of CLS molecules by forming Metal-O-S bonds [25]. Finally, LS^2−^ would adsorb on the outside film, forming a hydrophobic layer to protect the steel.

## 3. Materials and Methods

### 3.1. Materials and Test Solutions

The studied material was Q235 carbon steel with the following chemical composition (wt%): C 0.13, Si 0.29, Mn 0.76, S 0.11, Cu 0.08, and Fe balance. The sample size was 8 mm × 8 mm × 10 mm. The sample electrode was covered with epoxy resin, leaving a 0.4 cm^2^ area exposed to the test solution. The working surface for all samples was abraded with abrasive papers from 240# to 1000#, and cleaned with de-ionized water and ethanol.

The testing solution was composed of 0.0021 mol/L NaOH, 0.0042 mol/L KOH, and 0.02 mol/L NaCl, which was used to simulate the pore solution for a carbonated concrete environment [72]. The calcium lignosulfonate (CLS) and Na_2_MoO_4_ were analytical grade chemicals. The chemical structure of CLS is shown in Figure 14. The pH value of testing solutions was adjusted to 11.5, and all tests were carried out at room temperature (25 °C).

### 3.2. Electrochemical Measurements

Cyclic potentiodynamic polarization (CPP) curves and electrochemical impedance spectroscopy (EIS) were measured using CS350 electrochemical workstation (Corrtest Company, Wuhan, China). The polarization tests were started at a potential 300 mV below the open circuit potential (OCP) at a potential scanning rate of 0.1 mV/s in the anodic direction until the current density increased up to 0.02 mA/cm^2^, then the potential scanning was reversed at the same scanning rate until the test end. The corrosion potential (E_corr_) and corrosion current density (i_corr_) were obtained using Tafel plots. The inhibition efficiency (IE%) was calculated using Equation:(4)$$\mathrm{IE}\;(\%)=\frac{i_{corr}^{0}-i_{corr}}{i_{corr}^{0}}\times 100$$
where i^0^_corr_ and i_corr_ are the corrosion current density values in the test solutions without and with the inhibitors, respectively. In addition, the pitting potential (*E_b_*) is the critical potential above which pitting corrosion starts on steel surface [36], the repassivation potential (*E_pp_*) is the lowest potential for localized corrosion to propagate [38], and the pit transition potential (*E_ptp_*) is a characteristic potential that is correlated with the repassivation at the pit bottom [39,44,45].

The EIS measurements were performed under the corrosion potential with a potential perturbation of 15 mV and the scanning frequency range was 100 kHz‒0.01 Hz. The impedance data were fitted using ZSimpWin software. A three-electrode system was used in the electrochemical tests. A platinum electrode functioned as a counter electrode, the steel specimen worked as the working electrode, and the reference electrode was a saturated calomel reference electrode (SCE). Six parallel tests were run under each experimental condition.

An X-ray photoelectron spectroscopy (XPS) test was performed to analyze the surface composition with a Thermo Fisher ESCALAB 250 spectrometer (Qaltham, MA, USA). The samples were immersed in the carbonated SCP solution with different inhibitors for 24 h before XPS analysis. The binding energy values were calibrated using the C 1s peak at 248.8 eV.

To measure the wettability, the contact angle tests were performed on the inhibitor treated Q235 carbon steel surface using a JGW-36 contact angle test system (Chenghui Company, Chengde, China). During the test, the steel sample was mounted on the horizontal stage, the water drop (5 μL) was placed onto the steel surface using the syringe and the contact angle was measured using an integrated camera. For each sample, three distinct locations on the surface were measured and the average value was taken.

## 4. Conclusions

In carbonated SCP solution with 0.02 M NaCl, CLS could increase the pitting potential and decrease the corrosion current density for Q235 carbon steel, and the inhibition efficiency reaches 91.27% with 800 ppm CLS addition. Na_2_MoO_4_ also shows inhibition for both pitting and general corrosion, but the inhibition effect is obviously lower than that of CLS.

CLS and Na_2_MoO_4_ show a synergistic inhibition effect in the tested system. The compounded inhibitor can be defined as mix-type inhibitor. With 400 ppm CLS plus 600 ppm Na_2_MoO_4_, the pitting potential moves positively about 200 mV and the IE% reaches 92.67%. After 24 h immersion, the IE% further increases up to 99.2%.

XPS results confirm the adsorption of CLS and reveal the precipitation of oxides or hydroxides of Ca and Mo. Na_2_MoO_4_ could promote the stability of the passive film, and the insoluble molybdenum compounds and CaO/Ca(OH)_2_, together with the adsorbed CLS, deposit on the steel surface, forming a complex film. The compounded film effectively inhibits corrosion of the steel surface.

## Figures and Tables

**Figure 1 molecules-24-00518-f001:**
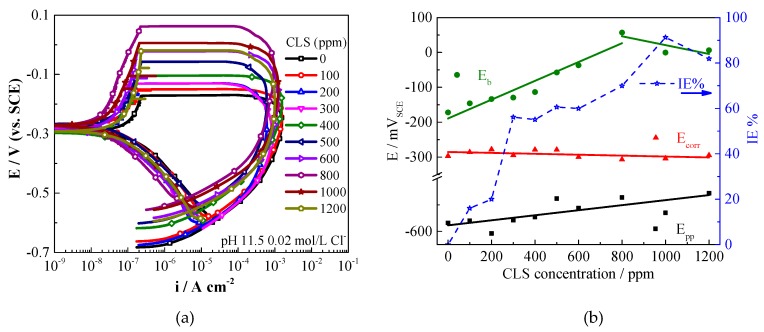
(**a**) Polarization curves of Q235 carbon steel in simulated concrete pore (SCP) solution with various concentration of calcium lignosulfonate (CLS); (**b**) Electrochemical parameters based on the polarization curves.

**Figure 2 molecules-24-00518-f002:**
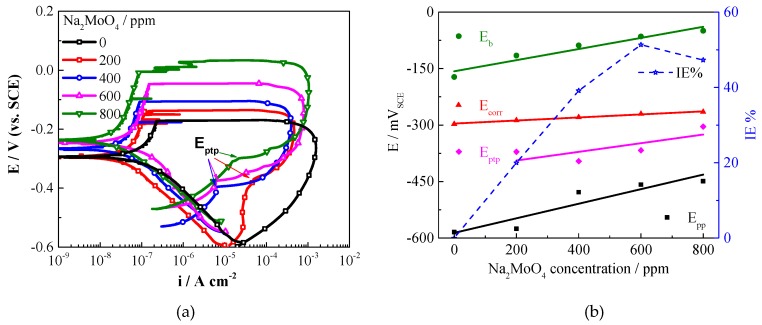
(**a**) Polarization curves of Q235 carbon steel in SCP solutions with various concentrations of Na_2_MoO_4_; (**b**) The electrochemical parameters based on the polarization curves.

**Figure 3 molecules-24-00518-f003:**
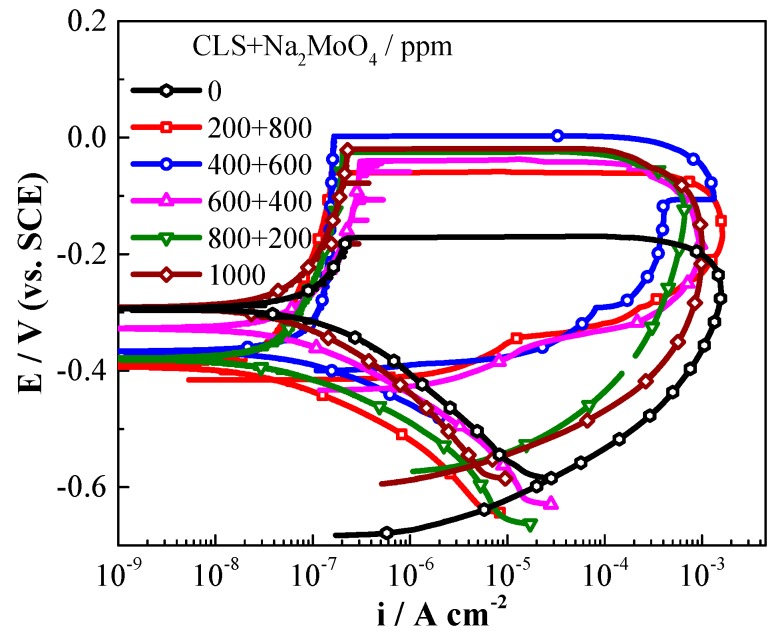
Polarization curves in solutions with different ratios of CLS and Na_2_MoO_4_.

**Figure 4 molecules-24-00518-f004:**
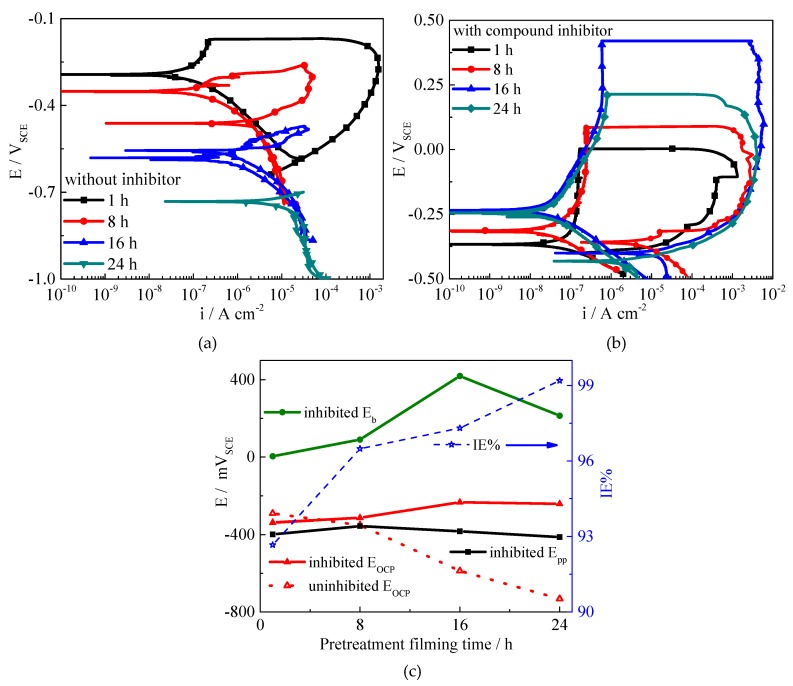
Polarization curves and parameters of Q235 steel immersed in carbonated SCP solution for various times, (**a**) without inhibitor, (**b**) with compound inhibitor, (**c**) electrochemical parameters obtained based on polarization curves.

**Figure 5 molecules-24-00518-f005:**
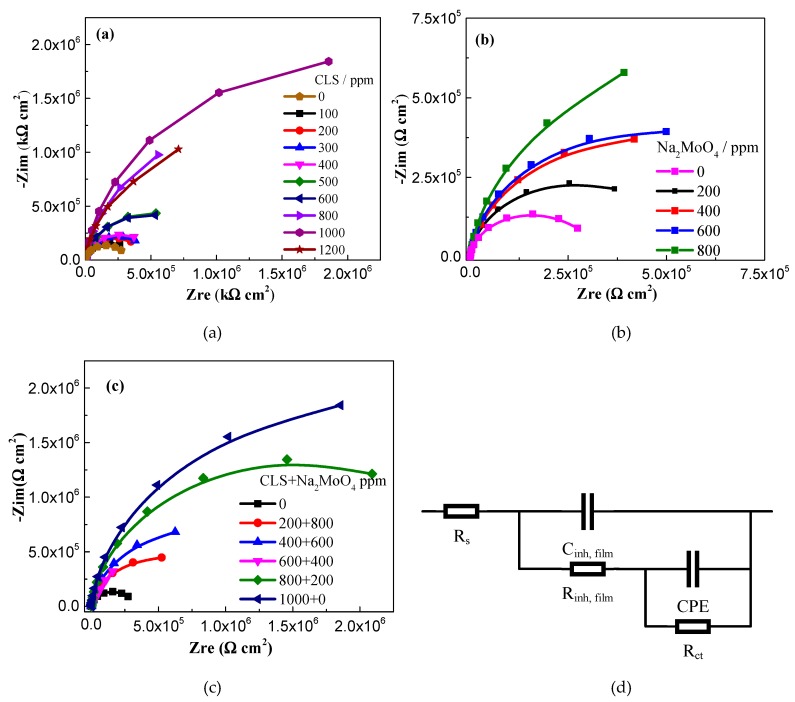
Nyquist plots of Q235 carbon steel in test solutions, (**a**) with CLS; (**b**) with Na_2_MoO_4_; (**c**) with optimal ratio of CLS and Na_2_MoO_4_; (**d**) the equivalent circuit.

**Figure 6 molecules-24-00518-f006:**
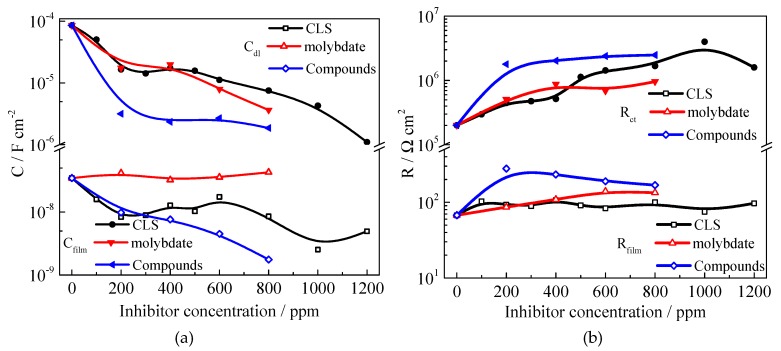
Electrochemical parameters of Q235 steel in carbonated SCP solution with different concentrations of inhibitor using electrochemical impedance spectroscopy (EIS) measurements, (**a**) C_dl_ and C_film_; (**b**) R_ct_ and R_film_.

**Figure 7 molecules-24-00518-f007:**
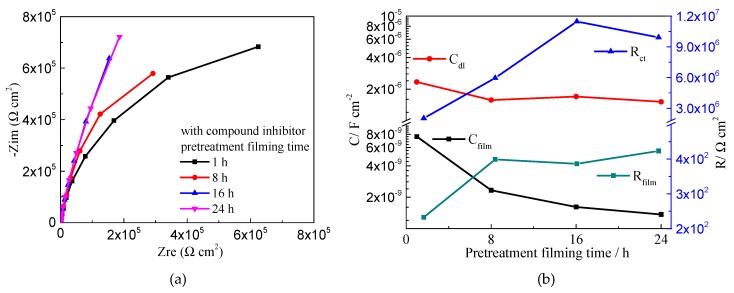
Nyquist polts (**a**) and EIS parameters (**b**) of Q235 steel in test solution with different pretreatment filming time.

**Figure 8 molecules-24-00518-f008:**
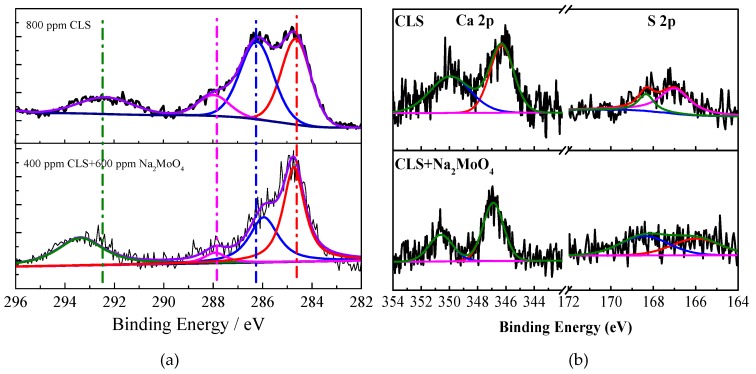
High-resolution XPS spectrum of Q235 steel immersed in test solution with CLS and CLS+Na_2_MoO_4_: (**a**) C 1s; (**b**) Ca 2p and S 2p.

**Figure 9 molecules-24-00518-f009:**
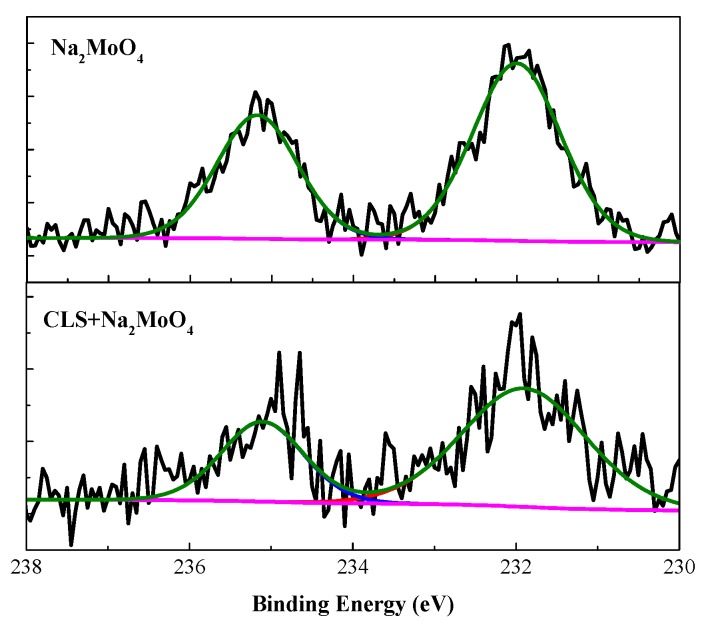
High-resolution XPS spectrum of Mo 3d for Q235 carbon steel immersed in carbonation SCP solution with Na_2_MoO_4_ and CLS+Na_2_MoO_4_.

**Figure 10 molecules-24-00518-f010:**
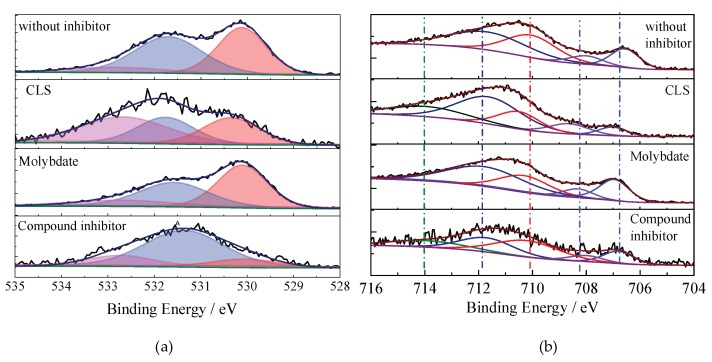
High-resolution XPS spectrum of Q235 carbon steel immersed in carbonation SCP solution with different inhibitors: (**a**) O 1s; (**b**) Fe 2p.

**Figure 11 molecules-24-00518-f011:**
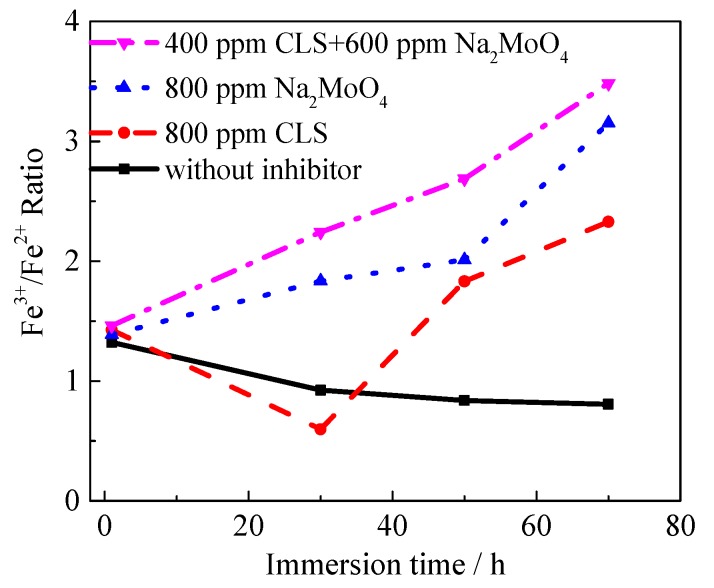
The Fe^3+^/Fe^2+^ ratio for Q235 steel immersed in test solution with different inhibitors.

**Figure 12 molecules-24-00518-f012:**
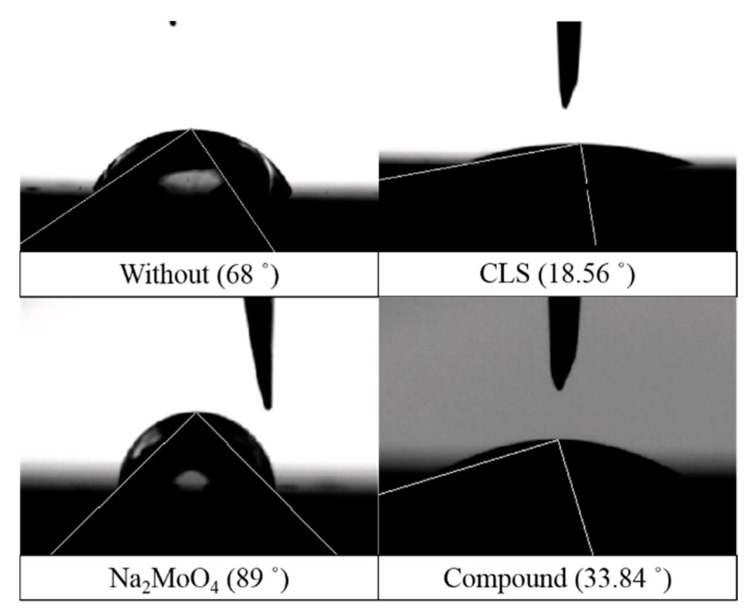
Typical images of water droplets on different inhibitor treated Q235 steel surfaces.

**Figure 13 molecules-24-00518-f013:**
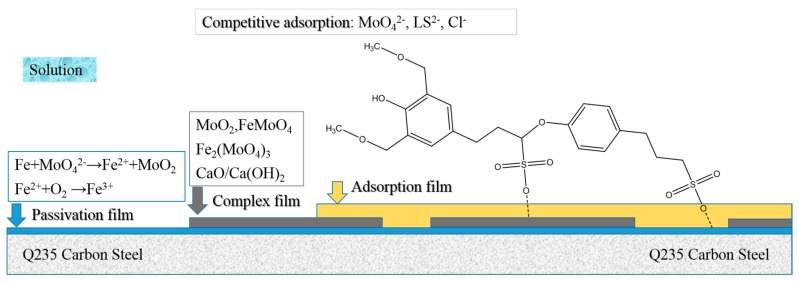
Illustration of the passivation-adsorption structure.

**Figure 14 molecules-24-00518-f014:**
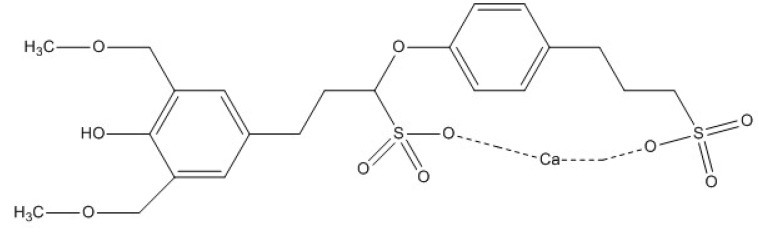
Chemical structure of calcium lignosulfonate (CLS).

**Table 1 molecules-24-00518-t001:** Electrochemical parameters of Q235 steel in carbonated SCP solution with different ratios of CLS and Na_2_MoO_4_ using cyclic potentiodynamic polarization (CPP) measurements.

Inhibitor		*E_corr_*	*E_b_*	*E_pp_*	IE%	S
CLS	Na_2_MoO_4_	mV_SCE_	mV_SCE_	mV_SCE_		
0	0	−298	−171.2	−584	−	−
200	800	−375	−73.2	−407	93.88	6.89
400	600	−338	3.9	−399	92.67	2.98
600	400	−310	−43.8	−429	81.73	1.34
800	200	−364	−15.3	−577	78.33	1.11
1000	0	−304	−0.3	−565	91.27	−

**Table 2 molecules-24-00518-t002:** Electrochemical parameters of Q235 steel in carbonated SCP solution with different ratios of CLS and Na_2_MoO_4_ using CPP measurements.

Inhibitor	Contact Angle (Degrees)			Average Contact Angle (Degrees)
	Location 1	Location 2	Location 3	
Without	68.00	66.75	71.54	68.76
CLS	18.32	18.56	16.37	17.75
Na_2_MoO_4_	85.84	89.00	79.04	84.63
Compound	30.12	33.84	36.42	33.46

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
