# Peer review of "Inhibition of Q235 Carbon Steel by Calcium Lignosulfonate and Sodium Molybdate in Carbonated Concrete Pore Solution"

_molecules, 2019, doi:10.3390/molecules24030518_

Round 1

Reviewer 1 Report

This paper shows interesting results for corrosion inhibition of Q235 carbon steel by CLS and Na2MoO4 with synergistic effect.

It could be acceptable for publication but need minor revisions.

Indeed, figures and equations are not properly numbered and a few other details should be revised (title, abbreviation, etc.).

See underline text with comments in joined pdf file.

Author Response

Dear Reviewer:

Manuscript ID: molecules-431003

Title: Inhibition of Q235 Carbon steel by Calcium Lignosulfonate and Sodium Molybdate in Carbonated Concrete Pore Solution

Thank you for your comments concerning our manuscript. Those comments are all valuable and very helpful for revising and improving our paper, as well as the important guiding significance to our researches. We have studied comments carefully and have made the corrections. We hope it meets with approval. 

Comments and Suggestions of reviewer:

This paper shows interesting results for corrosion inhibition of Q235 carbon steel by CLS and Na2MoO4 with synergistic effect. It could be acceptable for publication but need minor revisions. Indeed, figures and equations are not properly numbered and a few other details should be revised (title, abbreviation, etc.).

See underline text with comments in joined pdf file

Response: Thank you for your suggestions. We revised the figures and equations number, and other details were modified according to the suggestions of the reviewer. 

Reviewer 2 Report

Comments and Suggestions for Authors

Dear Authors,

I have to read your manuscript with great attention and interest. The material is consistent and valuable

The submission falls within the scope of the journal and is sufficiently original, and I have a remark, so I publish it after MINOR REVISIONS. MAJOR REMARKS: the manuscript should be rebuilt and the chapters changed.

General remarks:

Control the quality of Figures, make the bigger figures. Add the data of adhesion of layers and wettability. Wettability is very important properties take account corrosion. Add the adhesion of layers. Change the chapter “3. Materials and Methods” before “Discussion”. Present a test stand for electrochemical measurements

Keywords:

I suggest keyword: add “corrosion experiments”

Introduction:

Add some information about another inhibitors used for carbon steel.

Minor points:

12- explain SCP acronim

68- there isn’t any Fig. 1, It starts from Figures 2

75- b) electrochemical parameters obtained based on …

76- Ecorr- italic Ecorr

84- ipeak- italic

86- Epp- italic and the other variable parameters

116- check the equation number

169 from Fig a) by calculating the surface area (1st derived from the potentiokinetic curve), the ability to pass through the passive layer can be calculated

341- Figure 1

351- yellow place in equation

330 explain the corrosion examined parameters in this chapter and the corrosion experiments and station CHI, (reference electrode, etc, speed of sweep with potential) or put the scheme of the station for corrosive measurements

Author Response

Dear Reviewer:

Manuscript ID: molecules-431003

Title: Inhibition of Q235 Carbon steel by Calcium Lignosulfonate and Sodium Molybdate in Carbonated Concrete Pore Solution

Thank you for your comments concerning our manuscript. Those comments are all valuable and very helpful for revising and improving our paper, as well as the important guiding significance to our researches. We have studied comments carefully and have made the corrections. We hope it meets with approval. And the changes in the manuscript are as follows.

General remarks:

Control the quality of Figures, make the bigger figures. Add the data of adhesion of layers and wettability. Wettability is very important properties take account corrosion. Add the adhesion of layers. Change the chapter “3. Materials and Methods” before “Discussion”. Present a test stand for electrochemical measurements

Thank you for you suggest, we have resized the Figures to make it clearer. Add the data about wettability for Q235 carbon steel treated by different inhibitor. The adhesion of layers is an important parameter for inhibitor and we will try to understand that in the future.

Keywords:

I suggest keyword: add “corrosion experiments” 

Thank you for suggest to add keyword and we have added “corrosion experiments” in to the keywords.

Introduction:

Add some information about another inhibitor used for carbon steel.

Thank you for your suggest. We have introduced the traditional inorganic inhibitor in concrete in Introduction.

Minor points:

12- explain SCP acronim

We have added the explain of SCP.

68- there isn’t any Fig. 1, It starts from Figures 2

We checked and modified the number of Figures.

75- b) electrochemical parameters obtained based on …

We have modified the Figure caption.

76- Ecorr- italic Ecorr

84- ipeak- italic

86- Epp- italic and the other variable parameters

We modified the variable parameters in italics

116- check the equation number-

We have checked and modified the number of eqiation

169 from Fig a) by calculating the surface area (1st derived from the potentiokinetic curve), the ability to pass through the passive layer can be calculated

Thank you for your suggest and we will further calculate.

341- Figure 1

351- yellow place in equation

330 explain the corrosion examined parameters in this chapter and the corrosion experiments and station CHI, (reference electrode, etc, speed of sweep with potential) or put the scheme of the station for corrosive measurements

We have explained the corrosion examined parameters in the Chapter 3.

Reviewer 3 Report

This is an interesting paper and its conclusions are sound. However there are certain aspects that could be better clarified.  For example, the paragraph between lines 130 and 152 could be more clearly written and some of the observations require better explanations or should be left out..

Author Response

Dear Reviewer:

Manuscript ID: molecules-431003

Title: Inhibition of Q235 Carbon steel by Calcium Lignosulfonate and Sodium Molybdate in Carbonated Concrete Pore Solution

Thank you for your comments concerning our manuscript. Those comments are all valuable and very helpful for revising and improving our paper, as well as the important guiding significance to our researches. We have studied comments carefully and have made the corrections. We hope it meets with approval. 

Comments and Suggestions of reviewer:

This is an interesting paper and its conclusions are sound. However there are certain aspects that could be better clarified.  For example, the paragraph between lines 130 and 152 could be more clearly written and some of the observations require better explanations or should be left out.

Responds: Thank you for your suggestions. We rewrite the explanations of observations (between lines 130-152) as the suggest of reviewer.

Round 2

Reviewer 2 Report

Notes have been applied. I accept the submitted manuscript.